

# Version 1 NOAA-20/OMPS Nadir Mapper Total Column SO₂ Product: Continuation of NASA Long-term Global Data Record

Can Li[1], Nickolay A. Krotkov[1], Joanna Joiner[1], Vitali Fioletov[2], Chris McLinden[2], Debora Griffin[2], Peter J. T. Leonard[1,3], Simon Carn[4], Colin Seftor[1,5], Alexander Vasilkov[1,5]

[1]NASA Goddard Space Flight Center, Greenbelt, MD 20771, USA
[2]Environment and Climate Change Canada, Toronto, Ontario, Canada
[3]ADNET Systems, Inc., Lanham, MD 20706, USA
[4]Michigan Technological University, Houghton, MI 49931, USA
[5]Science Systems and Applications Inc., Lanham, MD 20706, USA

*Correspondence to*: Can Li (can.li@nasa.gov)

**Abstract.** For nearly two decades, the Ozone Monitoring Instrument (OMI) aboard the NASA Aura spacecraft (launched in 2004) and the Ozone Mapping and Profiler Suite (OMPS) aboard the NASA/NOAA Suomi National Polar-orbiting Partnership (SNPP) satellite (launched in 2011) have been providing global monitoring of SO₂ column densities from both anthropogenic and volcanic activities. Here, we describe the version 1 NOAA-20 (N20)/OMPS SO₂ product, aimed at extending the long-

term climate data record. To achieve this goal, we apply a principal component analysis (PCA) retrieval technique, also used for the OMI and SNPP/OMPS SO₂ products, to N20/OMPS. For volcanic SO₂ retrievals, the algorithm is identical between N20 and SNPP/OMPS and produces consistent retrievals for eruptions such as the 2018 Kilauea and 2019 Raikoke. For anthropogenic SO₂ retrievals, the algorithm has been customized for N20/OMPS, considering its greater spatial resolution and reduced signal-to-noise ratio as compared with SNPP/OMPS. Over background areas, N20/OMPS SO₂ slant column densities

(SCD) show relatively small biases, comparable retrieval noise with SNPP/OMPS (after aggregation to the same spatial resolution), and remarkable stability with essentially no drift during 2018-2023. Over major anthropogenic source areas, the two OMPS retrievals are generally well-correlated but N20/OMPS SO₂ is biased low especially for India and the Middle East, where the differences reach ~20% on average. The reasons for these differences are not fully understood but are partly due to algorithmic differences. Better agreement (typical differences of ~10-15%) is found over degassing volcanoes. SO₂ emissions

from large point sources, inferred from N20/OMPS retrievals, agree well with those based on OMI, SNPP/OMPS, and TROPOspheric Monitoring Instrument (TROPOMI), with correlation coefficients > 0.98 and overall differences < 10%. The ratios between the estimated emissions and their uncertainties offer insights into the ability of different satellite instruments to detect and quantify SO₂ sources. While TROPOMI has the highest ratios among all four sensors, ratios from N20/OMPS are slightly greater than OMI and substantially greater than SNPP/OMPS. Overall, our results suggest that the version 1

N20/OMPS SO₂ product will successfully continue the long-term OMI and SNPP/OMPS SO₂ data records. Efforts currently underway will further enhance the consistency of retrievals between different instruments, facilitating the development of multi-decade, coherent global SO₂ datasets across multiple satellites.



## 1 Introduction

Sulfur dioxide ($SO_2$) is emitted from both anthropogenic (*e.g.*, burning of sulfur-containing fuels, oil and gas exploitation, and
metal smelting) and volcanic activities. The adverse impacts of $SO_2$ and its secondary product in the atmosphere, sulfate
aerosols, have been well documented, including their effects on air quality (*e.g.*, Wang et al., 2016), visibility (*e.g.*, Hand and
Malm, 2007), human health (*e.g.*, Orellano et al., 2021, and references therein), and ecosystems (*e.g.*, Fedkin et al., 2019;
Likens et al., 1996). By scattering solar radiation (*e.g.*, Chuang et al., 1997) and acting as cloud condensation nuclei (*e.g.*,
Haywood and Boucher, 2000), sulfate aerosols also influence earth's radiation budget and the climate. Explosive volcanic
eruptions inject sizable amounts of $SO_2$ and other species (*e.g.*, water vapor, halogens, $CO_2$) into the stratosphere, leading to
significant perturbations of stratospheric aerosols (*e.g.*, Vernier et al., 2011) and consequently, substantial impacts on the
global climate (*e.g.*, Aubry et al., 2021; McGraw et al., 2024; Robock, 2000, Stenchikov, 2016; Timmreck, 2012) as well as
stratospheric ozone (*e.g.*, Evan et al., 2023; Solomon et al., 1996; Zhu et al., 2018). To better understand the influence of
volcanic $SO_2$ on the earth system, it is imperative to develop and maintain global monitoring capabilities from satellites, given
that volcanic eruptions often happen with little warning in remote areas. In addition, large and varying changes in
anthropogenic $SO_2$ emissions have occurred across different regions in recent decades (*e.g.*, Krotkov et al., 2016), amid various
factors such as economic development, energy structure, and environmental policies. Long-term, global $SO_2$ datasets are also
valuable for the detection and attribution of these changes, offering insights into, for example, the efficacy of pollution control
measures.

Polar-orbiting satellites equipped with spectrometers that measure back-scattered solar radiation in the ultraviolet (UV) have
been a major asset for global $SO_2$ monitoring. Heritage instruments such as the Total Ozone Mapping Spectrometer (TOMS)
take measurements at a small number of wavelengths and can only detect relatively large amounts of $SO_2$ (*e.g.*, Krueger, 1983;
Fisher et al., 2019), but provide a record of $SO_2$ from major eruptions dating back to the late 1970s (Carn, 2022). First launched
in the 1990s, UV/Visible spectrometers (*e.g.*, GOME, the Global Ozone Monitoring Experiment) make measurements at
hundreds of wavelengths, allowing detection of $SO_2$ signals from degassing volcanoes and large anthropogenic sources
(Eisinger and Burrows, 1998). More recent instruments with 2-dimensional detectors such as the Ozone Monitoring Instrument
(OMI, Levelt et al., 2018) and the TROPOspheric Monitoring Instrument (TROPOMI, Veefkind et al., 2012) are capable of
daily global observations at greater spatial resolution, enhancing sensitivity to smaller $SO_2$ emission sources (e.g., Krotkov et
al., 2006; Theys et al., 2015; Yang et al., 2007).

Along with advances in instrumentation, progress in retrieval techniques has led to continued improvements in satellite data
products. Data-driven methods, for example the principal component analysis (PCA) based algorithm (Li et al., 2013) and the
COvariance Based Retrieval Algorithm (COBRA, Theys et al., 2021), have proved to be useful for $SO_2$ retrievals. Both PCA
and COBRA $SO_2$ algorithms inherently account for various interferences as well as instrumental factors. As a result, they can
produce $SO_2$ retrievals with reduced noise and biases as compared with other methods such as Differential Optical Absorption



Spectroscopy (DOAS), band residual difference, and linear fit. The data-driven retrieval technique is also relatively insensitive to drift in instrument calibration, thus helping to maintain the stability of long-term data record. For example, NASA's latest standard OMI $SO_2$ product based on the PCA algorithm shows little change in the mean $SO_2$ over background areas during the 15-year period between 2004 and 2019 (Li et al., 2020). The PCA $SO_2$ algorithm has also been implemented with the Ozone Mapping and Profiler Suite (OMPS) Nadir Mapper (NM) aboard the NASA/NOAA Suomi National Polar-orbiting Partnership

(SNPP) spacecraft. Despite the relatively coarse spatial ($50 \times 50$ km$^2$ at nadir) and spectral resolution (~1 nm vs. ~0.6 nm) of SNPP/OMPS as compared with OMI, the PCA-based retrievals are largely consistent between the two instruments for both anthropogenic (Zhang et al., 2017) and volcanic (Li et al., 2017) $SO_2$.

An important application enabled by this progress in satellite instruments and retrieval techniques is to use satellite data to infer $SO_2$ emissions from large point sources (Fioletov et al., 2015). These top-down estimates, compiled in a publicly available

catalogue (Fioletov et al., 2016), offer independent constraints on annual emissions from both anthropogenic sources (Li et al., 2017; Liu et al., 2018; McLinden et al., 2021; Zhang et al., 2019) and degassing volcanoes (Carn et al., 2017), and have helped to uncover emission sources that were previously missing from bottom-up inventories (McLinden et al., 2016). The most recent $SO_2$ emission catalogue (version 2, Fioletov et al., 2023) has been updated to include inputs from multiple satellite instruments including OMI, SNPP/OMPS, and TROPOMI.

With OMI approaching the end of its mission by 2026 and SNPP/OMPS already in its second decade of operation, data products from newer instruments are needed to continue the long-term $SO_2$ climate data record. The inclusion of TROPOMI in the emission catalogue is a necessary first step, but the existing TROPOMI $SO_2$ products (Theys et al., 2017; 2021) are not fully consistent with NASA's OMI/OMPS products due to differences in algorithms and ancillary datasets. In addition, the follow-on instruments for TROPOMI are planned for morning orbits (Fig. 1) that may pose additional challenges from a data

continuity perspective. Like TROPOMI, the U.S. Joint Polar Satellite System 1 (JPSS-1, also known as NOAA-20)/OMPS was also launched in 2017 and has similar spatial resolution as OMI. The four OMPS instruments planned for the JPSS program (including the two already launched on NOAA-20 and NOAA-21) can potentially extend global $SO_2$ monitoring capabilities from the afternoon orbits into the 2040s (Fig. 1). Our goal here is to produce a continuity NOAA-20 (N20)/OMPS $SO_2$ product that bridges SNPP/OMPS with the follow-on JPSS/OMPS sensors. To this end, we have implemented the PCA-based $SO_2$

algorithm with N20/OMPS. In this paper, we describe our version 1 N20/OMPS $SO_2$ product that has been publicly released (Li et al., 2023). The rest of the paper is organized as follows: section 2 briefly introduces the PCA $SO_2$ retrieval algorithm and specific implementation details for N20/OMPS. It also provides a description of the data product files. In section 3, we assess the quality of N20/OMPS $SO_2$ retrievals and compare them with SNPP retrievals. This is followed by data availability statement in section 4 and conclusions in section 5.

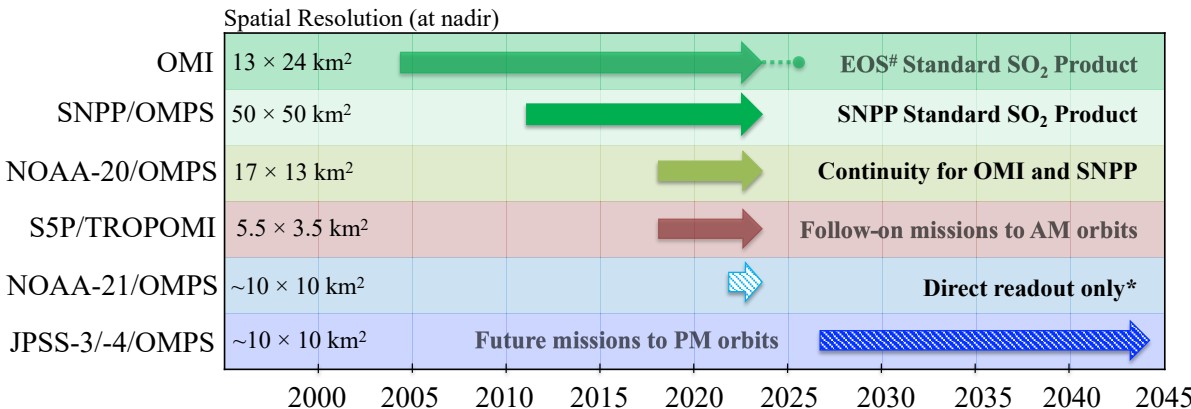

#EOS: Earth Observing System. *As of January 2024.


**Figure 1: Current and planned UV instruments for global monitoring of SO₂ from the afternoon Sun-synchronous orbits.**

## 2 Data and Methodology

### 2.1 NOAA-20/OMPS Nadir Mapper

The NOAA-20 (N20) spacecraft is the first of four satellites planned for the NASA/NOAA JPSS program and flies in a Sun-synchronous, ascending orbit with a local equator crossing time of approximately 1:30 p.m. Flying in the same orbit plane as SNPP, N20 initially operated ahead of SNPP by a half orbit (~50 min) and currently leads SNPP by a quarter orbit (~25 min), after the launch of NOAA-21. The N20/OMPS instrument suite is comprised of two nadir-viewing spectrometers (the Nadir Profiler (NP) and Nadir Mapper (NM)), whereas its predecessor, SNPP/OMPS, also contains a limb-viewing spectrometer (the Limb Profiler, LP). The N20/OMPS NM sensor measures earthshine radiances and solar irradiance in the wavelength range of 300–420 nm at a spectral resolution of ~1 nm. With a 110° field of view (FOV), it covers a cross-track swath of ~2800 km and provides nearly daily global coverage. The configurable 2-D Charge Coupled Device (CCD) detector of N20/OMPS NM contains 340 pixels in the spectral dimension and 720 pixels in the spatial (cross-track) dimension (Wang et al., 2022). At the beginning of the mission, the pixels in the spatial dimension were aggregated to 104 macropixels (cross-track positions or CCD rows). This configuration, together with a 2.5 second integration time along the flight direction, provides a spatial resolution of 17 km (along-track) × 17 km (across-track) at nadir. Starting from orbit 6419 on 13 February 2019, the spatial pixels have been aggregated to 140 macropixels, providing a finer spatial resolution of 13 km across-track at nadir. As compared with its counterpart on SNPP, N20/OMPS NM has a spatial resolution that is ~10 times greater, but its signal-to-noise ratio is lower by approximately a factor of 3-4. The enhanced spatial resolution allows N20/OMPS to detect smaller SO₂ sources, but also leads to larger retrieval noise as shown in the following sections of the paper.



## 2.2 PCA-based SO₂ retrieval algorithm

Detailed descriptions of our PCA-based SO₂ retrieval algorithm and its implementation with OMI and SNPP/OMPS have been given elsewhere (Li et al., 2013, 2017, 2020). Here we provide a brief overview of the algorithm (section 2.2.1) and the modifications that are specific to implementation with N20/OMPS (section 2.2.2).

### 2.2.1 Algorithm overview

Our PCA SO₂ retrieval algorithm utilizes satellite measured, sun normalized earthshine radiances at the top of the atmosphere (TOA) in the spectral range of ~310-340 nm. By applying a PCA technique to the radiance spectra, we extract spectral features (principal components, or PCs) and rank them based on the spectral variance they each explain. In the absence of large SO₂ signals (*e.g.*, from volcanic eruptions), the leading PCs (that explain the most spectral variance) are often associated with geophysical processes (*e.g.*, ozone absorption, rotational Raman scattering) or instrument measurement details (*e.g.*, wavelength shift, dark current) other than SO₂ absorption in the atmosphere. We then fit the first $n_v$ none-SO₂ PCs ($v_i$), along with the absorption cross sections of SO₂ ($\sigma_{SO_2}$), to the measured TOA radiances ($I$), to simultaneously estimate the coefficients of the PCs ($\omega_i$) and the SO₂ slant column density ($SCD_{SO_2}$).

$$-ln\big[I\big(\omega, SCD_{SO_2}\big)\big] = \sum_{i=1}^{n_v} \omega_i v_i + SCD_{SO_2}\sigma_{SO_2} \,. \tag{1}$$

The number of PCs ($n_v$) included in Eq. (1) is usually set at 20-30 depending on the number of wavelengths in the fitting window but can be smaller if potential SO₂ features are identified in the leading PCs due to unscreened SO₂ signals in the radiance spectra (see Li et al., 2013 for details). The SO₂ SCD is then converted to vertical column density (VCD, $\Omega_{SO_2}$) using an air mass factor (AMF):

$$\Omega_{SO_2} = \frac{SCD_{SO_2}}{AMF} \,. \tag{2}$$

The AMF is calculated using radiative transfer (RT) code that accounts for factors such as the sun-target-satellite geometry, surface albedo and surface pressure, cloud fraction and cloud pressure, ozone amount and profile, temperature profile, as well as the *a priori* profile of SO₂. Alternatively, we can also fit the radiance spectra using the PCs and the total column Jacobians of SO₂ ($\frac{\partial ln(I)}{\partial \Omega_{SO_2}}$) that are determined from RT calculations and represent the sensitivity of TOA radiances at different wavelengths to a perturbation in SO₂ VCD. This allows us to obtain VCDs in a single step.

For implementation with OMI and SNPP/OMPS, we process data from each cross-track position (or row) of the 2-D detector separately, effectively treating each as an individual spectrometer. To minimize the impact of orbit-to-orbit changes in the measured radiances (for example in dark current), we also process each orbit separately. As a result, the input data (for each





row of a given orbit) to the PCA algorithm typically include ~1600 radiance spectra (or pixels) sampled along the flight track of OMI (~400 spectra for SNPP/OMPS). They are subject to three successive processing steps:

Step 1) Initial data screening: the spectra are first screened to exclude pixels with large solar zenith angles (SZA > 75°) or
those potentially affected by the South Atlantic anomaly (SAA). They are then examined for potential large volcanic $SO_2$ signals by computing the residuals (*i.e.*, the differences between the measured and calculated TOA radiances) at two wavelength pairs (313/314 nm and 314/315 nm), using $O_3$ column amounts from the total $O_3$ product (Bhartia, 2005) and the simple Lambertian equivalent reflectivity (SLER, Ahmad et al. 2004) derived at longer wavelengths (342, 354, and 367 nm). The radiance calculations assume zero $SO_2$, and the wavelength pairs are chosen to detect the spectral contrast between
wavelengths near (313 and 315 nm) and off (314 nm) the $SO_2$ absorption peaks. Spectra that have relatively large residuals at 313 and 315 nm as compared with 314 nm are considered to potentially contain large $SO_2$ signals (*e.g.*, from a volcanic plume) and are excluded from the PCA analysis, although $SO_2$ retrievals are still conducted for those pixels (see Li et al., 2020 for details).

Step 2) PCA analysis and additional $SO_2$ screening: after filtering large volcanic $SO_2$ signals, we attempt to remove any residual
$SO_2$ signals in the remaining radiance spectra, using two procedures. In the first procedure, a PCA analysis is conducted on the spectra and the resulting leading PCs are used to fit the spectra, essentially reconstructing the spectra using those PCs. We flag pixels as potentially $SO_2$ contaminated, if they have residuals that are spectrally correlated with the $SO_2$ cross sections (see Li et al. 2020). This helps to filter out relatively small $SO_2$ signals, as compared with the volcanic $SO_2$ screening in Step 1). In the second procedure, we conduct PCA on the remaining spectra and use the PCs and $SO_2$ cross sections (or Jacobians)
to obtain first guess $SO_2$ retrievals (Eq. 1). Pixels having relatively large negative ($< -2\sigma$ from the mean, where $\sigma$ is the standard deviation) or positive ($> 1.5\sigma$ from the mean) first guess $SO_2$ are excluded from additional PCA analyses. We divide the pixels from the row into 3 subgroups based on their latitudes and for each subgroup, we repeat the second procedure twice to derive the final PCs and $SO_2$ SCDs.

Step 3) Jacobian calculations and final $SO_2$ VCD estimates: using the final PCs from Step 2) and Jacobians calculated
employing a table lookup approach, we conduct spectral fitting to obtain final estimates of $SO_2$ VCDs for all pixels that have SZA ≤ 75°, including those flagged for $SO_2$ during Steps 1) and 2). The Jacobian calculations are separate for anthropogenic and volcanic $SO_2$ retrievals. For anthropogenic $SO_2$ retrievals, the Jacobians are calculated once for each pixel using cloud fraction and cloud pressure from the rotational Raman scattering cloud product (Joiner and Vasilkov, 2006; Vasilkov et al., 2014) and *a priori* $SO_2$ profiles based on a climatology from multi-year global model simulations (see Li et al., 2020 for
details). For volcanic $SO_2$ retrievals, particularly for explosive eruptions, the $SO_2$ Jacobians strongly depend on the $SO_2$ amounts, and absorption signals at shorter wavelengths may become saturated. To account for these factors, we use an iterative procedure to estimate volcanic $SO_2$ Jacobians based on the retrieved $SO_2$ VCD from the previous iteration and optimize the fitting window by dropping potentially saturated wavelengths (see Li et al., 2017 for details). As the volcanic $SO_2$ plume





heights are often unknown immediately following eruptions, in our volcanic $SO_2$ retrievals we also produce four estimates of
the $SO_2$ VCD for each pixel, assuming four different *a priori* profiles. These $SO_2$ profiles centered at 3, 8, 13, and 18 km
altitudes, respectively, are chosen to represent typical plume heights from volcano degassing (3 km: lower troposphere, TRL),
moderate eruptions (8 km: middle troposphere, TRM), or explosive eruptions (13 km: upper troposphere, TRU and 18 km:
lower stratosphere, STL).

**2.2.2 Implementation with NOAA-20/OMPS**

While our N20/OMPS $SO_2$ algorithm shares the same general design with our OMI and SNPP/OMPS algorithms, some
implementation details differ. Here, we summarize the algorithm modifications that are specific to the current version of the
N20/OMPS $SO_2$ algorithm.

Volcanic $SO_2$ screening: for N20/OMPS, instead of the residual-based scheme (see section 2.2.1), we have implemented a new
scheme based on spectral fitting using reference PCs from a presumably $SO_2$-free orbit over the remote Pacific. For orbits with
104 rows (see section 2.1), the reference PCs are derived on a row-by-row basis from orbit 4506 on 1 October 2018, whereas
for orbits with 140 rows, the reference PCs are from orbit 17460 on 1 April 2021. Both are days without major volcanic
eruptions and the leading modes of the reference PCs are likely free from $SO_2$ features. We conduct spectral fits using the
reference PCs and a $SO_2$ Jacobian spectrum assuming 18-km plume height to obtain an initial $SO_2$ VCD estimate for each
pixel. Pixels having initial $SO_2 > 2$ DU (Dobson Units, 1 DU = $2.69 \times 10^{16}$ molecules cm$^{-2}$) are flagged for potential volcanic
influence.

Anthropogenic $SO_2$ retrievals: other than the scheme for initial volcanic $SO_2$ screening, the algorithm for volcanic $SO_2$
retrievals as implemented with N20/OMPS is identical to that for SNPP/OMPS. The N20/OMPS anthropogenic $SO_2$ algorithm,
on the other hand, has several changes. These include:

1) Algorithm settings for SCD retrievals: for N20/OMPS $SO_2$ SCD retrievals, pixels within each row are grouped into five
subsectors, instead of three as for OMI and SNPP/OMPS (see section 2.2.1). Additionally, in step 2) of the N20 algorithm,
pixels having first guess $SO_2$ that falls within $\pm 1.5\sigma$ from the mean are considered $SO_2$-free and retained for additional PCA
analysis. This is different from OMI and SNPP/OMPS algorithms that retain pixels with first guess $SO_2$ between (mean - $2\sigma$)
and (mean + $1.5\sigma$). The change in the threshold was made to mitigate potential positive biases in $SO_2$ SCDs, although this may
have led to an overall negative bias for N20/OMPS $SO_2$ SCDs as compared with SNPP (see section 3). Additionally, for OMI
and SNPP/OMPS, an iterative process is used to examine fitting residuals for SCD retrievals over areas affected by the South
Atlantic Anomaly (SAA) and to exclude wavelengths that have large residuals in a second step SCD fitting. This process helps
to reduce SCD noise over the SAA areas but has not yet been implemented with N20/OMPS.





2) AMF/Jacobians: in the current version of the N20/OMPS anthropogenic $SO_2$ algorithm, a fixed AMF (0.36) is used to convert all SCDs to $SO_2$ VCDs, regardless of the observation conditions for different pixels. This AMF corresponds to a

simplified scenario under cloud-free conditions with fixed surface albedo (0.05) and pressure (1013.25 hPa), solar (30°) and viewing (0°) zenith angles, and typical mid-latitude temperature and $O_3$ profiles (total column $O_3$ = 325 DU). It is also assumed that $SO_2$ is mostly in the planetary boundary layer (PBL), or the lowest ~1 km of the atmosphere. The same AMF was used for $SO_2$ VCDs, referred to as PBL $SO_2$, in the early versions of OMI $SO_2$ product (Krotkov et al., 2006). Following this convention, we use "ColumnAmountSO2_PBL" as the data field name for $SO_2$ VCDs derived using this fixed AMF in our

version 1 N20/OMPS $SO_2$ product (see section 2.3 for description). We plan to produce a refined anthropogenic $SO_2$ VCD dataset using the same Jacobian calculation method as for OMI and SNPP/OMPS (see section 2.2.1), once the N20/OMPS Raman cloud product, currently under development, becomes available.

3) Algorithmic refinement for pixels in the transition zones between different subsectors: it has been previously noted that there are relatively large gradients in OMI $SO_2$ SCD uncertainties (and to a lesser extent, in SCDs) near the boundaries between

different subsectors, as different sets of PCs are used in spectral fitting for those subsectors (see Figure 2, Li et al., 2020). To reduce this gradient, we have refined N20/OMPS SCD retrievals for pixels within the transition zone, defined here as the 50 pixels located immediately across the boundary between two subsectors (with 25 on either side). For each transition zone pixel, we conduct multiple spectral fits using different sets of PCs from all subsectors. We then select the fit that has the smallest root mean square (RMS) of the fitting residuals for the final $SO_2$ SCD.

**2.3 Description of version 1 NOAA-20/OMPS $SO_2$ product**

Detailed description of the current version of our PCA-based N20/OMPS $SO_2$ product (product name: OMPS_N20_NMSO2_PCA_L2_Step1), including its file format in netCDF-4 and data fields, is given in the product readme file, available at https://disc.gsfc.nasa.gov/datasets/OMPS_N20_NMSO2_PCA_L2_Step1_1/summary (Li et al., 2023). A summary of the data fields that are of interest to most data users is given below.

**SlantColumnDensitySO2:** $SO_2$ slant column densities (SCDs) in molecules cm$^{-2}$ derived using sun-normalized radiances between 310.5 and 340 nm.

**SLER**: simple Lambertian equivalent reflectivity (Ahmad et al. 2004) at the terrain pressure at three wavelengths, 342, 354, and 367 nm.

**ColumnAmountSO2_PBL:** $SO_2$ vertical column densities (VCDs) in Dobson Units estimated using $SO_2$ SCDs and a fixed

AMF of 0.36 (see section 2.2). The AMF is calculated assuming that $SO_2$ is predominantly in the boundary layer, hence the name for the data field. The PBL $SO_2$ data can be used for studies on anthropogenic $SO_2$ pollution, but pixels with large SLER (for example > 0.2 at 342 nm) should be excluded to reduce the cloud effects that are unaccounted for in the AMF.





**ColumnAmountSO2_TRL:** $SO_2$ VCDs (DU) estimated assuming a lower troposphere (TRL) *a priori* $SO_2$ profile with a center mass altitude of 3 km. The TRL $SO_2$ data can be used for studies on volcanic degassing.

**ColumnAmountSO2_TRM:** $SO_2$ VCDs (DU) estimated assuming a middle troposphere (TRM) *a priori* $SO_2$ profile with a center mass altitude of 8 km. The TRM $SO_2$ data can be used for moderate volcanic eruptions.

**ColumnAmountSO2_TRU:** $SO_2$ VCDs (DU) estimated assuming an upper troposphere (TRU) *a priori* $SO_2$ profile with a center mass altitude of 13 km. The TRU $SO_2$ data can be used for studying explosive eruptions that inject $SO_2$ into the upper troposphere.

**ColumnAmountSO2_STL:** $SO_2$ VCDs (DU) estimated assuming a lower stratosphere (STL) *a priori* $SO_2$ profile with a center mass altitude of 18 km. The STL $SO_2$ data can be used for studies on explosive eruptions that directly inject $SO_2$ into the lower stratosphere.

### 2.4 Multi-satellite $SO_2$ emission catalogue

Our previous version 2 global catalogue of large $SO_2$ emission sources is based on OMI, SNPP/OMPS, and TROPOMI data,
covers the period of 2005-2021, and includes a total of 759 continuously emitting point sources releasing from about 10 kt $y^{-1}$ to more than 4000 kt $y^{-1}$ of $SO_2$ (Fioletov et al., 2023). Here, we use N20/OMPS $SO_2$ data to estimate annual emissions for these sources and then use the N20 emission estimates, in addition to those from the other three satellite sensors, to produce an updated (2005-2023) unified emission catalogue. For N20/OMPS emission estimates, we apply the the same algorithm as for the other satellite sensors, including the same site-specific AMFs. For each source, we first estimate a local retrieval bias
based on the average upwind $SO_2$ that is subsequently subtracted from the $SO_2$ retrievals. To estimate emissions, the total average $SO_2$ mass near the source is calculated using a fitting algorithm and then emissions are derived as the ratio of the total mass to the lifetime, assumed to be constant at 6 h. In addition to the assumed lifetime, the algorithm uses a prescribed constant parameter ($\omega$) that represents the average plume width across the wind direction. The values of the prescribed parameter are $\omega$=20 km, 25 km, and 15 km for OMI, SNPP/OMPS and TROPOMI, respectively. The parameter $\omega$=20 km is chosen for
N20/OMPS as it has similar pixel sizes as OMI. As in Fioletov et al. (2023), the OMI and OMPS-based emission estimates presented here have been increased by +10 % to match the values to the earlier version of the catalogue (Fioletov et al., 2016). Similarly, for TROPOMI, a +22% correction is applied to account for differences in temperatures for the $SO_2$ absorption cross sections used in the retrievals.

### 3 Results and Discussion

In this section, we evaluate our version 1 N20/OMPS $SO_2$ product. Given the similarities between the N20 and SNPP/OMPS NM sensors, we focus on the comparisons between the two OMPS $SO_2$ products. Sect. 3.1-3.3 are dedicated to analyses of $SO_2$ SCDs and PBL $SO_2$; we first assess the quality and stability of N20/OMPS $SO_2$ SCDs (Sect. 3.1) and then compare the spatial distribution (Sect. 3.2) and long-term time series (Sect. 3.3) of N20 PBL $SO_2$ retrievals with those from SNPP. In Sect.





3.4, we present the results using N20/OMPS SO$_2$ SCD data for emission estimates and compare N20-based emission estimates

with other instruments. In Sect. 3.5, we compare N20 and SNPP/OMPS volcanic SO$_2$ retrievals for selected eruptions.

### 3.1 Quality and long-term stability of NOAA-20/OMPS SO$_2$ slant column densities

In Fig. 2, we compare the statistics of N20 and SNPP/OMPS SO$_2$ SCDs over the east Pacific on 1 April 2019. For such a day without major volcanic eruptions, the actual loading and variability of SO$_2$ are presumed to be quite small over remote background areas, and the mean and standard deviation of the retrieved SO$_2$ SCDs can be used to assess the biases and noise

in the retrievals. As shown in Fig. 2a, both mean SNPP and N20/OMPS SO$_2$ SCDs are within ±0.05 DU, indicating relatively small biases for both retrievals, although N20/OMPS SCDs are in general smaller than those from SNPP/OMPS, especially at middle (30-50°N) to high (70-80°N) latitudes in the northern hemisphere. For the standard deviation of SCDs (Fig. 2b), both N20 and SNPP/OMPS show dependence on latitude, likely due to a generally smaller signal-to-noise ratio (SNR) in the measurements at lower radiance values associated with larger solar zenith angles (SZAs). Apart from the latitude band of 70-

80°N, the N20/OMPS SCD standard deviation is ~0.35-0.65 DU at its native resolution, or approximately 2-4 times greater than that of SNPP/OMPS. Recall that the size of each SNPP/OMPS pixel is ~10 times larger than N20/OMPS, due to the longer integration time along track (7.5 s versus 2.5 s) and the aggregation of more pixels into fewer rows (macropixels) across track (36 versus 140 rows in this example). This indicates that the SCD retrieval noise from the two OMPS instruments approximately scales according to $\sqrt{N}$, where $N$ is the number of aggregated pixels, suggesting that the greater noise in

N20/OMPS SO$_2$ SCDs is largely driven by larger random noise in the radiances. Indeed, if we average the N20/OMPS SO$_2$ SCDs to the same resolution as SNPP/OMPS, the standard deviation of binned SCDs (blue triangles, Fig. 2b) is now comparable with that of SNPP/OMPS at most latitudes. The one noticeable exception is at 70-80°N, indicating additional SCD retrievals errors at larger SZAs for N20/OMPS. Note that for this analysis, the mean of SZAs for 70-80°N is similar between the two OMPS instruments. For analyses in Sect. 3.2-3.3, pixels with SZA > 65° are excluded.

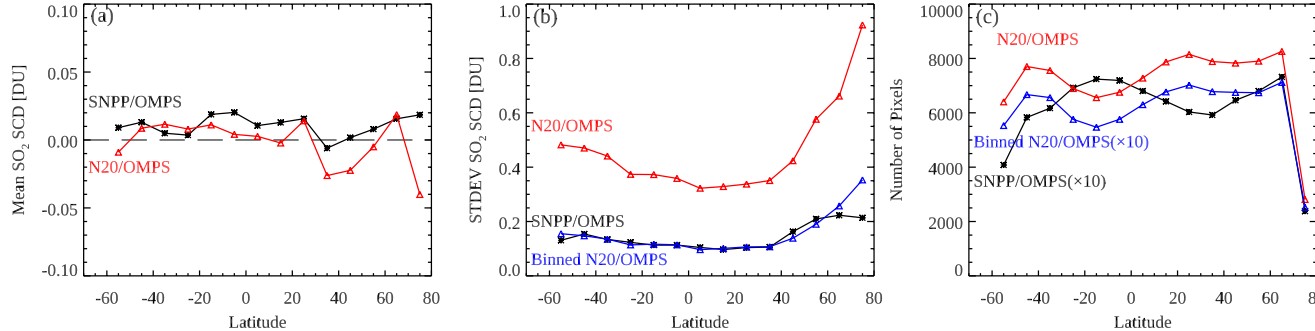


**Figure 2: Comparisons of the (a) mean, (b) standard deviation, and (c) number of pixels for SO$_2$ SCD retrievals for 10° latitude bands over the remote Pacific (130-150°W) on 1 April 2019 between SNPP/OMPS (black asterisks), N20/OMPS at its native resolution (red triangles), and N20/OMPS binned to the same resolution as SNPP/OMPS (blue triangles, for standard deviation and**


**pixel number only). Results are given in Dobson Units (DU). All pixels that have solar zenith angle < 70° are included in the analysis.**

**Only latitude bands having at least 500 native N20/OMPS pixels from the day are shown. On this day, N20 was operating a half orbit (~50 min) ahead of SNPP, and the number of SNPP and binned N20/OMPS pixels is different but comparable for all latitudes. The smaller number of pixels at high latitudes is due to the limit on the solar zenith angle.**

For long-term monitoring, it is important to minimize the drift over time that can introduce spurious trends in the dataset. To evaluate the stability of OMPS $SO_2$ retrievals, we examine the daily mean and standard deviation of $SO_2$ SCDs over the

equatorial Pacific, after screening out days affected by large volcanic eruptions. As can be seen from Fig. 3a, both N20 and SNPP/OMPS retrievals are quite stable, showing no statistically significant trends in the mean $SO_2$ SCDs from the beginning of the missions through 2023. The change in the SNPP/OMPS SCD standard deviation is also very small at 0.0002 DU y$^{-1}$ (Fig. 3b). One may notice a jump in the standard deviation for N20/OMPS SCDs (red, Fig. 3b) in early 2019 due to the change in instrument spatial resolution (Sect. 2.1), but the trend is again quite small at -0.00023 DU y$^{-1}$, once N20/OMPS SCDs are

binned to the same resolution as SNPP/OMPS. The changes for both OMPS instruments are much smaller than a previous study on OMI $SO_2$ retrievals by Li et al. (2020), who reported trends of 0.00023 DU y$^{-1}$ and 0.0015 DU y$^{-1}$ in the mean and standard deviation of OMI SCDs, respectively, over the equatorial Pacific during 2004-2019.

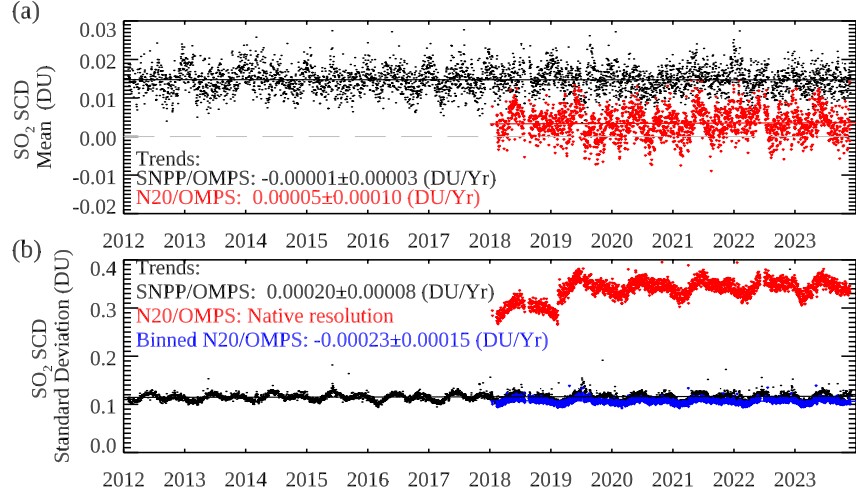

**Figure 3: Time series of daily (a) mean and (b) standard deviation of $SO_2$ SCDs over the equatorial Pacific (20°S-20°N, 130-150°W)**

**from N20 and SNPP/OMPS from the beginning of the data records to 2023. In panel (b) results at native N20/OMPS resolution (red) and those aggregated to the SNPP/OMPS resolution (blue) are given to account for the change in instrument resolution in February 2019. Days affected by volcanic plumes are excluded, as are days with fewer than 500 pixels within the domain. For N20/OMPS retrievals at native resolution, the first and last two rows are excluded due to relatively large retrieval noise. The estimated trends from linear regression are provided along with their 95% confidence intervals, except for the standard deviation at N20/OMPS**

**native resolution.**





## 3.2 Spatial distribution of PBL SO$_2$

Figure 4 presents the spatial distribution of PBL SO$_2$ VCDs from N20 and SNPP/OMPS and their differences for March to
May 2021. The period has been selected due to its lack of major volcanic eruptions to facilitate comparisons for anthropogenic
sources and degassing volcanoes. It is worth mentioning that the analysis here reflects the differences in SCDs between the
two OMPS products, as the same fixed AMF (0.36, see Sect. 2.2.2) is applied everywhere. Overall, the spatial distribution of
PBL SO$_2$ VCDs is quite similar between the two OMPS products, with both showing relatively small values over oceanic areas
and hotspots over major anthropogenic sources (*e.g.*, Norilsk in Russia, South Africa, northeast India, and Persian Gulf) as
well as degassing volcanoes (*e.g.*, Kilauea in Hawaii, Nyiragongo in Congo, and Krakatau in Indonesia). More analyses on the
long-term PBL SO$_2$ time series over selected source areas, as marked in Figs. 4a and 4b, are given in Sect. 3.3. There are also
noticeable differences (Fig. 4c). Over the SAA-affected areas, N20/OMPS PBL SO$_2$ VCDs have larger noise that is partly due
to the algorithmic difference in SCD fitting for the region (see Sect. 2.2.2). For most other areas in the middle and low latitudes,
N20/OMPS VCDs are smaller, particularly over the dust belt from the Sahara to northwestern China and Mongolia. As
discussed in Sect. 2.2.2, at least some of these negative biases in N20/OMPS retrievals (as compared with SNPP) can be
attributed to the algorithm settings in SCD fitting, namely the threshold (based on first guess SO$_2$ estimates) used to filter out
potentially SO$_2$-contaminated pixels. Over most areas, the negative bias is relatively small. For example, only ~1% of the grid
cells over the dust belt (defined here as the domain of 20-50°N, 0-110°E) have VCD differences that exceed -0.2 DU (*i.e.,* -
0.072 DU in SCD difference). Despite their relatively small magnitude, the biases in N20/OMPS retrievals can still lead to
substantial differences in the regional PBL SO$_2$ time series (see Sect. 3.3 for details), but they can be well mitigated in the top-
down emission estimates (Sect. 3.4). Over high latitudes, near the coastal areas of Greenland, Antarctica, and the Arctic,
N20/OMPS PBL SO$_2$ VCDs also have relatively large positive and negative biases (Fig. 4c). This is consistent with the results
shown in Fig. 2 and points to potential algorithm issues for scenes with low SNRs, although the exact reason is unknown as
of the writing of this manuscript.






**Figure 4: Mean PBL SO₂ VCDs for March to May 2021 retrieved from (a) SNPP/OMPS and (b) N20/OMPS using a fixed AMF**

**(0.36) and (c) their differences. Both retrievals are gridded to 0.5°×0.5° horizontal resolution, and N20/OMPS data are binned to the**

**same spatial resolution as SNPP before gridding to ensure consistent sampling. For both SNPP and the binned N20 datasets, pixels**

**with SZA > 65° or those from the extreme off-nadir rows (first two and last two) are excluded. There is no data screening based on**

**cloudiness. Blue rectangular boxes in (a) and (b) mark the domains of selected anthropogenic SO₂ source areas, whereas the green**

**boxes mark selected degassing volcanoes.**

The scatter plot (Fig. 5a) between the two gridded datasets over the entire global domain (excluding the SAA region) indicates

that the N20 and SNPP/OMPS PBL SO₂ VCDs are moderately correlated. Most (~81.8%) of the grid cells have near zero

values (within ±0.1 DU) from both retrievals. This is as expected since SO₂ loading outside of source areas is quite small in

the absence of large volcanic plumes. About 19.6% of the grid cells have absolute VCD differences > 0.1 DU, likely reflecting

the differences between the two retrievals especially over background areas and at high latitudes, although the absolute

differences exceed 0.2 DU for only ~2.7% of them. For grid cells with SNPP/OMPS PBL SO₂ VCDs > 1 DU, 234 (out of 248)

have N20 and SNPP/OMPS VCDs agreeing to within ±30%. This is also demonstrated by the scatter plot in Fig. 5b that focuses

on the selected source areas (as marked by boxes in Figs. 4a and 4b.). For these areas, N20 and SNPP/OMPS PBL SO₂ VCDs

are strongly correlated ($r = 0.98$) and have a small overall bias (slope = 1.03), suggesting overall good consistency between

the two instruments for relatively strong SO₂ signals.

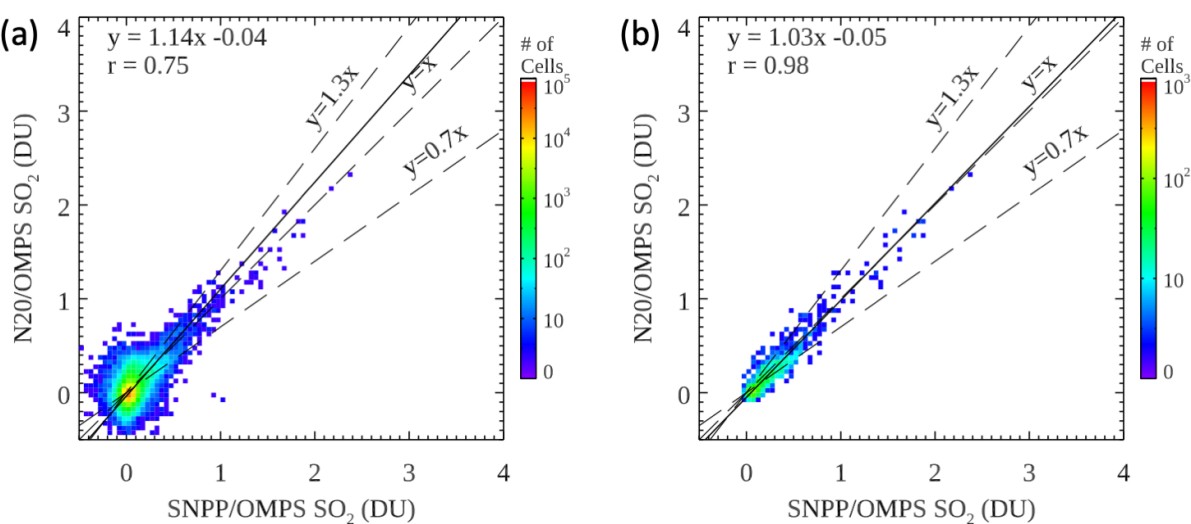


**Figure 5: (a) The density map of gridded PBL SO₂ VCDs from N20/OMPS *vs.* SNPP/OMPS for the same period and same domain**

**(excluding SAA affected areas) as in Fig. 4. Colors represent the number of 0.5°×0.5° grid cells. The solid black line marks the best**

**fit line through all grid cells from a linear regression analysis. The dashed lines represent scenarios where N20 SO₂ is 30% higher**

**than (y = 1.3x), equal to (y = x), and 30% lower than (y = 0.7x) SNPP, respectively. The correlation coefficient ($r = 0.75$) between**





**N20 and SNPP/OMPS VCDs for all grid cells is also given. (b) Same as (a) but only for selected source areas as marked by rectangular boxes in Figures 4a and 4b.**

### 3.3 Long-term time series of PBL SO$_2$

To evaluate the consistency between N20 and SNPP/OMPS PBL SO$_2$ VCDs over extended periods, we also compare the time series of monthly mean SO$_2$ mass (in kiloton, $10^3$ metric tonnes) over selected major source areas (see Fig. 4a and Fig. 4b for

the domains for anthropogenic and volcanic sources, respectively). For a given area, we first generate daily gridded N20 and SNPP/OMPS PBL SO$_2$ VCDs at 0.5°×0.5° resolution following the same procedure and data filtering criteria as described for Fig. 4. For each day when over 90% of the domain is covered by the gridded data, we calculate the total SO$_2$ mass by summing up the mass from all grid cells that have non-negative gridded SO$_2$ VCDs. The monthly mean SO$_2$ mass is then calculated by averaging the daily data.

In addition, we attempt to correct for the monthly and latitude dependent biases in N20/OMPS PBL SO$_2$. We first produce monthly gridded N20 and SNPP/OMPS PBL SO$_2$ VCDs from the daily gridded data. For each month of the year, we use the monthly gridded data to estimate the latitude dependent biases in three steps: 1) filtering out areas that have relatively large SO$_2$ (monthly SNPP/OMPS PBL VCD > 0.5 DU) or those affected by SAA; 2) calculating the mean N20-SNPP differences within 3° latitude bands for the same month of each year during 2018-2023; and 3) taking the median of the monthly mean

biases from the 6-year period for each latitude band. The estimated biases are then used to produce bias-corrected N20 PBL SO$_2$ times series (blue lines in Figs. 6 and 7).

For anthropogenic source areas, the N20 and SNPP/OMPS PBL SO$_2$ time series are mostly well-correlated (Fig. 6). The lowest correlation coefficient ($r$ = 0.64) is found over Mt. Isa in Australia, where the SO$_2$ mass is typically below 1 kt (Fig. 6e). For all other areas, the correlation coefficient exceeds 0.8. The 12-year SNPP/OMPS data record reveals some significant regional

trends in SO$_2$ pollution. For instance, SO$_2$ over India gradually increases over time; there is a temporary dip in 2020 probably related to the COVID-19 pandemic (e.g., Biswas and Ayantika, 2021) followed by increases afterwards. Meanwhile, SO$_2$ over China has decreased substantially since 2014, likely due to emission control measures (Li et al., 2017b). Qualitatively, these long-term regional changes are also confirmed by the N20/OMPS time series. On average, the N20/OMPS SO$_2$ mass over Persian Gulf, India, China, and South Africa is smaller than SNPP/OMPS by -20.1%, -17.6%, -13.4%, and -13.4%,

respectively. In comparison, the differences are much smaller over Norilsk and Mt. Isa at 4.7% and -0.6%, respectively. The monthly latitudinal bias correction improves the correlation between the N20 and SNPP/OMPS time series for all regions (Fig. 6), but the remaining differences between the two are still substantial over Persian Gulf (-15.0%), India (-10.6%), and South Africa (-9.5%), indicating marginal improvements for these aeras. The differences between the N20 and SNPP/OMPS are little changed over Norilsk (8.0%) and Mt. Isa (-1.5%) after the bias correction. The results here suggest that the simple latitude

dependent correction as outlined above is insufficient to significantly improve the agreement between the two datasets everywhere.

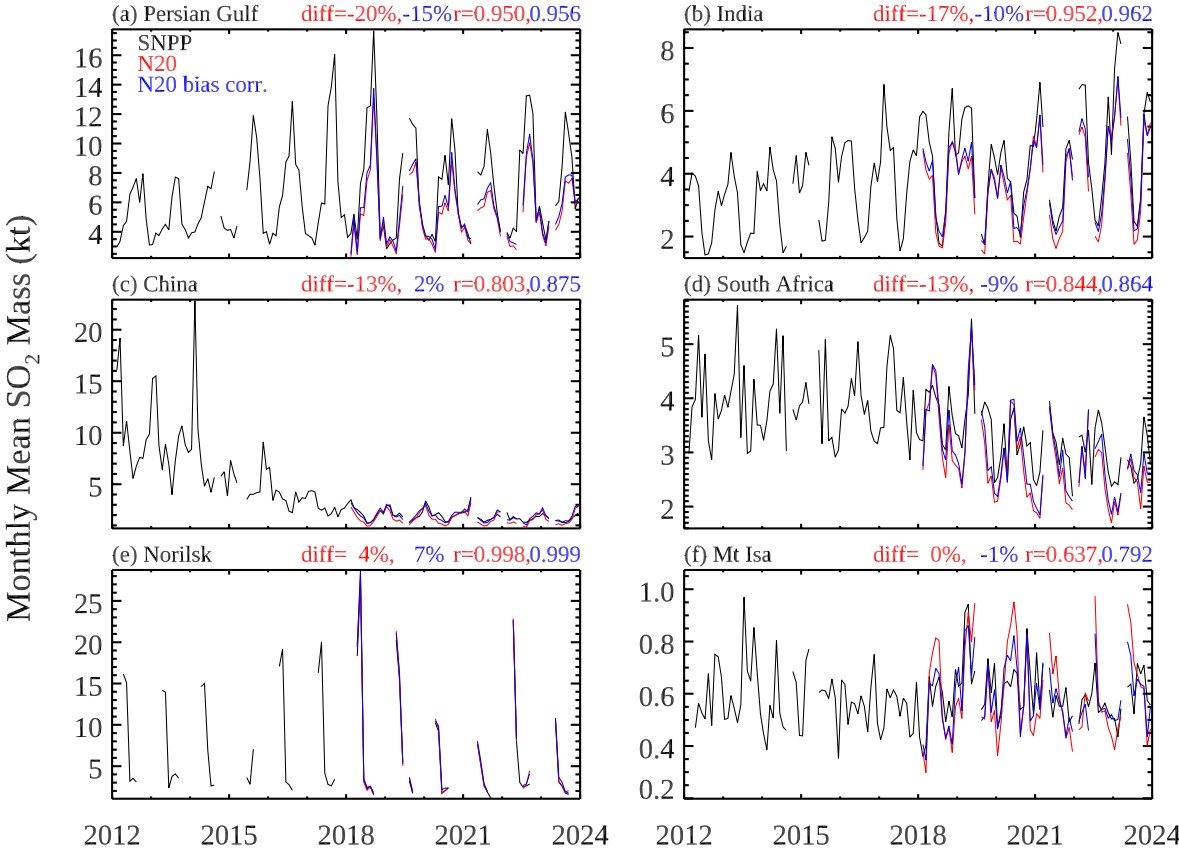

**Figure 6: Time series of monthly mean SO₂ mass (in kiloton, 10³ tonnes) based on SNPP/OMPS (black), N20/OMPS (red), and bias corrected N20/OMPS (blue) PBL SO₂ VCDs over major anthropogenic source areas as marked in Fig. 4a. For a given area, only months having at least 10 days' worth of data are shown. Months influenced by large volcanic SO₂ plumes (September 2014, April to May 2015, July 2019, April 2021, January 2022, and April 2023) are also excluded. The average percentage differences and correlation coefficients between N20 and SNPP/OMPS time series (red), as well as those between the bias corrected N20 and SNPP/OMPS time series (blue) are given at the top of each panel. Note that the large seasonal change over Norilsk is likely due to snow/ice that is currently unaccounted for in retrievals.**

As for degassing volcanoes, the PBL SO₂ time series in Fig. 7 show strong correlation between the two OMPS products. The correlation coefficient exceeds 0.9 for all cases and could be attributed to the large variations in the volcanic SO₂ emissions as well as generally consistent retrievals. Before the bias correction, the average relative differences between N20 and SNPP/OMPS during 2018-2023 are -12.2%, -10.6%, -24.8%, and 8.2% for Popocatepetl, Kilauea, Nyiragongo, and Sabancaya and Ubinas, respectively. The large relative difference over Nyiragongo is probably caused by the small SO₂ loading since late 2021, whereas the positive bias over Sabancaya and Ubinas could be due to the greater impact of SAA on N20/OMPS retrievals. After the bias correction, and the average relative differences are -8.0%, 0.1%, -11.7%, and 8.3% for the four areas,

respectively. It should be pointed out that for degassing volcanoes, the PBL SO$_2$ retrievals likely overestimate SO$_2$, as the fixed AMF used in these retrievals represents a scenario with SO$_2$ predominantly in the boundary layer, while SO$_2$ plumes from the volcanoes are typically at higher altitudes. In Sect. 3.5, we compare N20 and SNPP/OMPS volcanic SO$_2$ retrievals that use more representative *a priori* SO$_2$ profiles and Jacobians calculated for individual pixels (see Sect. 2.2.1).

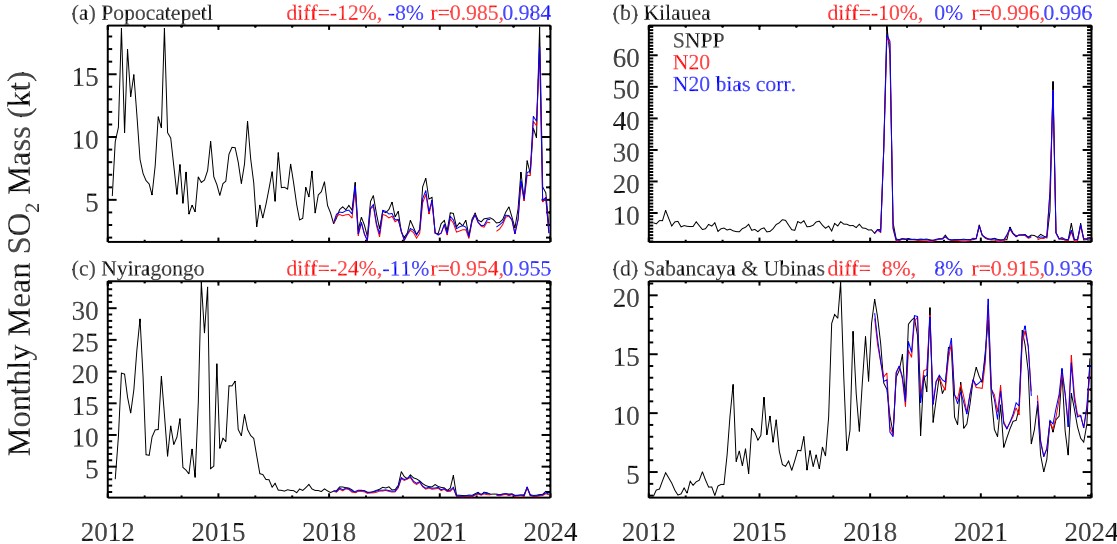

**Figure 7: Same as Figure 6 but for degassing volcanoes (as marked in Fig. 4b) and all months during 2012-2023 (*i.e.*, no exclusion of months affected by large volcanic eruptions).**

### 3.4 SO$_2$ Emission estimates for large point sources

To assess the ability of N20/OMPS to detect and quantify SO$_2$ point sources, we apply a top-down emission estimation algorithm (Fioletov et al., 2015, 2016, 2023) to N20/OMPS retrievals to derive annual emissions from 759 large point sources that are included in the version 2 satellite-based SO$_2$ emission catalogue (Fioletov et al., 2023). As evidenced by Fig. 8, there is strong correlation between emissions estimated using N20/OMPS and those using OMI (Fig. 8a), SNPP/OMPS (Fig. 8b), and TROPOMI (Fig. 8c), with correlation coefficients > 0.98 in all cases. The overall biases are also quite small with agreement within ~10% for all sensors: the estimated total annual emissions for all the sources, averaged over the period of 2018-2023, are 44.7, 47.0, 49.1, and 45.7 Mt y$^{-1}$ (Megaton, $10^6$ tonnes) for N20/OMPS, OMI, SNPP/OMPS, and TROPOMI, respectively. Note that the top-down emission algorithm relies on relative enhancement in SO$_2$ signals over a relatively small domain (see section 2.4). This may explain the overall small differences in the estimated emissions between N20 and SNPP/OMPS (~10%), despite the substantial differences in their PBL SO$_2$ column density time series as discussed in Sect. 3.3.





Figures 8d, 8e, and 8f compare the ratios between the estimated emissions and their associated uncertainties for different instruments. The uncertainties in the estimated emissions depend on the number of available retrievals as well as their noise. For the largest sources, all four sensors can well capture the enhancement in $SO_2$, and the ratios are quite similar between different instruments. For smaller sources, TROPOMI has the best overall sensitivity and greatest ratios (Fig. 8f) owing to its fine spatial resolution and large number of retrievals. The ratios for N20/OMPS are overall comparable with those for OMI

(Fig. 8d), with the former having slightly larger ratios for most sources. N20/OMPS provides more retrievals than OMI, but its advantage in sample size is likely partially offset by the relatively large retrieval noise. As for the comparison between N20 and SNPP/OMPS (Fig. 8e), N20/OMPS has greater ratios for most of the sources. The results here imply that given two similarly designed instruments, the one with higher spatial resolution but larger noise will likely offer a stronger capability for point source detection and quantification.

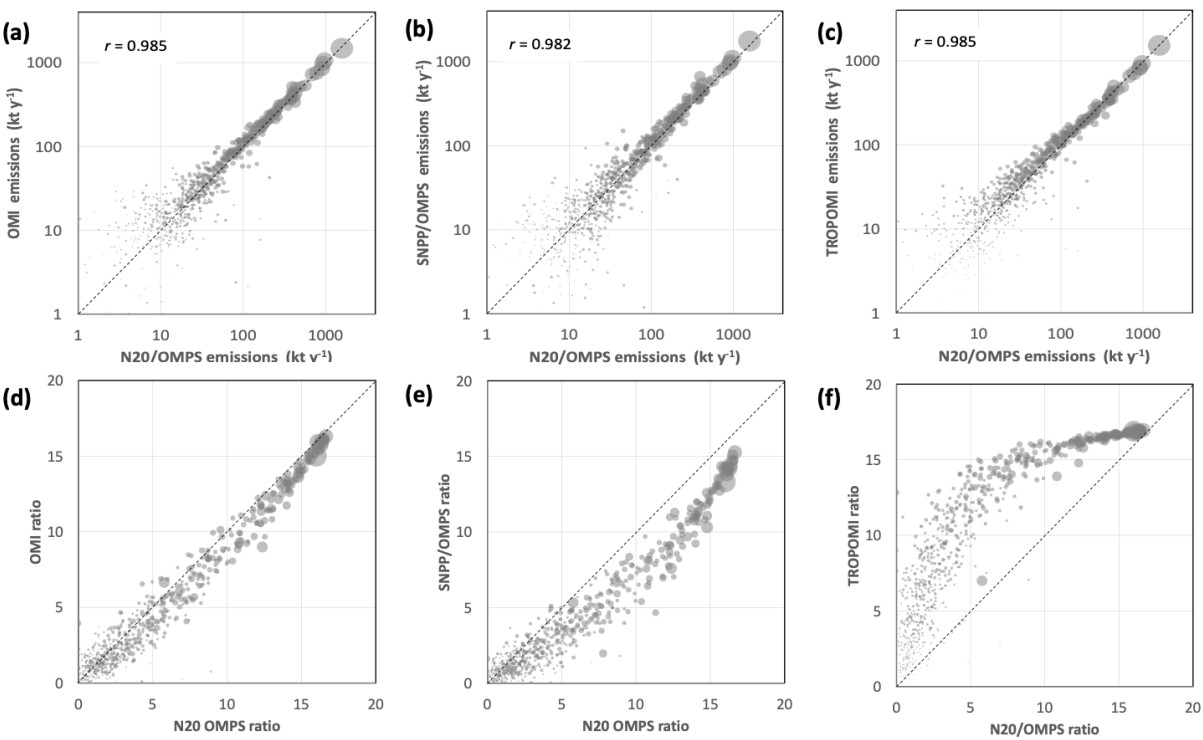


**Figure 8: Upper: scatter plots comparing the annual emissions of $SO_2$ (kt y$^{-1}$) averaged over 2018-2023 for large point sources (in the version 2 emission catalogue) based on $SO_2$ retrievals from (a) OMI *vs.* N20/OMPS, (b) SNPP/OMPS *vs.* N20/OMPS, and (c) TROPOMI *vs.* N20/OMPS. Lower: scatter plots of the ratios between the estimated emissions and their uncertainties based on retrievals from (d) OMI *vs.* N20/OMPS, (e) SNPP/OMPS *vs.* N20/OMPS, and (f) TROPOMI *vs.* N20/OMPS. The dashed lines are**

**1:1 lines. Each bubble represents a point source, and the bubble area is proportional to the annual $SO_2$ emissions from the source.**





Figure 9 presents the time series of regional $SO_2$ emissions during 2005-2023, based on emission estimates from the four individual satellite sensors. Over the period of 2018-2023, N20/OMPS shows similar changes in $SO_2$ emissions as other instruments. For example, the emissions from Europe and the U.S. are relatively steady after significant declines in the 2000s and 2010s. Emissions from China have continued to decrease, although at a much lower rate than between 2014 and 2017.

Emissions from India and the Middle East saw a drop around 2020 and have since recovered. The N20/OMPS based emissions also generally agree with other instruments for most regions, although relatively large differences are found for India and the Middle East, especially during 2018-2020 when the N20/OMPS based emissions are ~10-15% smaller than the weighted average. SNPP/OMPS based emissions for these two regions, on the contrary, appear to be biased high. It is possible that the relatively large differences in PBL $SO_2$ between the two OMPS instruments (see Sect. 3.3) may have been caused by the

negative biases in N20/OMPS retrievals as well as the positive biases in SNPP/OMPS retrievals.

Using the emissions derived separately from individual satellites, we obtain the unified emission estimates for the catalogue by calculating a weighted average of the emission estimates from the four satellite instruments using an inverse-variance weighting method. The weighted averages are also shown in Fig. 9 (the red lines). In version 2 catalogue, OMI, SNPP/OMPS, and TROPOMI data contribute 7%, 5%, and 88% to the average respectively for small (< 30 kt $y^{-1}$) sources and 33%, 20%,

and 47% respectively for large (> 300 kt $y^{-1}$) sources. The addition of N20/OMPS changes the contribution of individual instruments to 6%, 4%, 10%, and 80% for small sources and 23%, 15%, 26%, and 35% for large sources (for OMI, SNPP/OMPS, N20/OMPS, and TROPOMI, respectively). Thus, the weighting coefficients for N20/OMPS are nearly twice of those for SMPP/OMPS. Again, this suggests that the superior spatial resolution of N20/OMPS is beneficial for emission estimates and yields a substantially lower uncertainties in emissions than SNPP/OMPS, despite overall similar retrieval noise

between the two when binned to the same spatial resolution (see Fig. 2b).

Overall, these emission estimates demonstrate that N20/OMPS has the potential to augment and further extend the long-term satellite based $SO_2$ emission catalogue.

Data

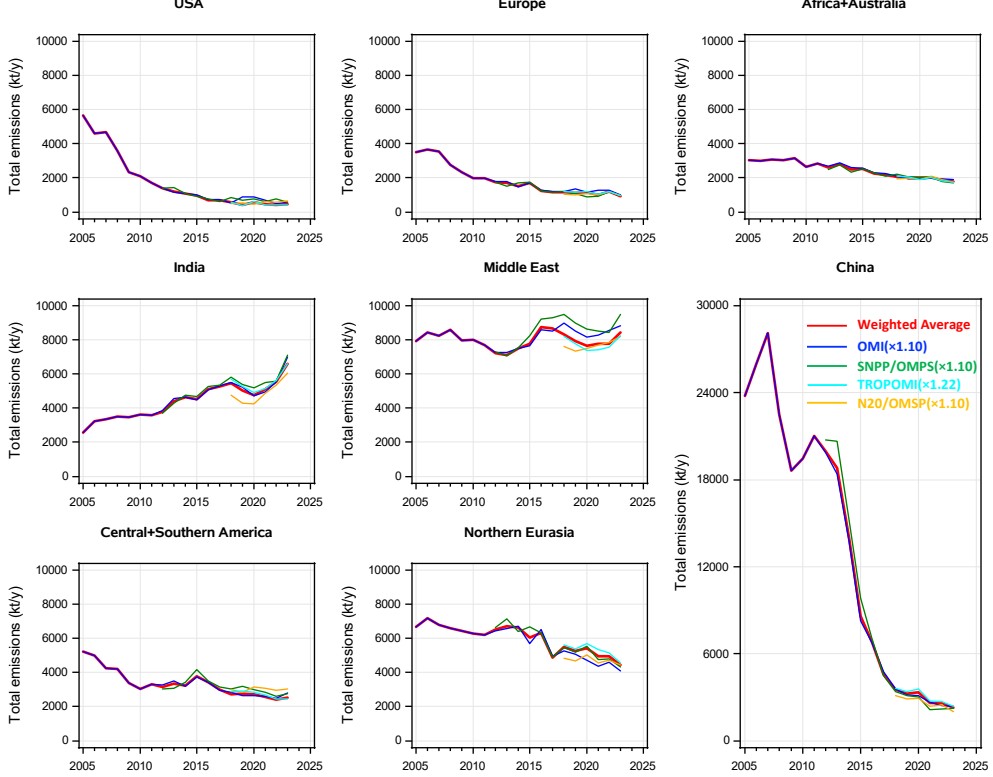

**Figure 9: Annual emissions for different regions during 2005-2023 from the point sources that are included in the version 2 satellite**
**based SO₂ emission catalogue. Colors represent emission estimates using different satellite sensors, as well as their weighted average.**
**For individual satellite datasets, the scaling factors have been applied to match with an earlier version of the catalogue or to account**
**for differences in SO₂ cross sections in retrievals (see section 2.4 and Fioletov et al., 2023 for details).**

### 3.5 Comparisons of volcanic SO₂ retrievals between SNPP and NOAA-20

In this section, we compare volcanic SO₂ VCDs between N20 and SNPP/OMPS for two eruptions that are quite different in
terms of their strength and SO₂ emissions.

The first case for comparison is the fissure eruption of Kilauea volcano that started on 3 May 2018 and lasted for a few months.
SO₂ released from this eruption remained low in the troposphere (Tang et al., 2020), and we compare N20 and SNPP/OMPS
TRL (lower troposphere) SO₂ retrievals that assume an *a priori* profile centered at 3 km altitude (see Sect. 2.3). Given that the
same algorithm is used for N20 and SNPP/OMPS volcanic SO₂ retrievals (Sect. 2.2.2), one would expect generally consistent
results between the two. Indeed, as shown in Fig. 10c, the estimates of SO₂ mass over the domain around the volcano, calculated
daily from N20 and SNPP/OMPS retrievals for the period of May-July 2018, are well-correlated with $r > 0.9$. The overall bias
between the two datasets is also relatively small (slope = 1.05, intercept = 0.09 from a linear regression analysis). On the other

hand, the differences between N20 and SNPP/OMPS mass estimates can exceed 10 kt at times. This is likely due to the very different pixel size of the two instruments (see Figs. 10a and 10b for an example) that in some cases may lead to sampling

biases for a relatively small domain.

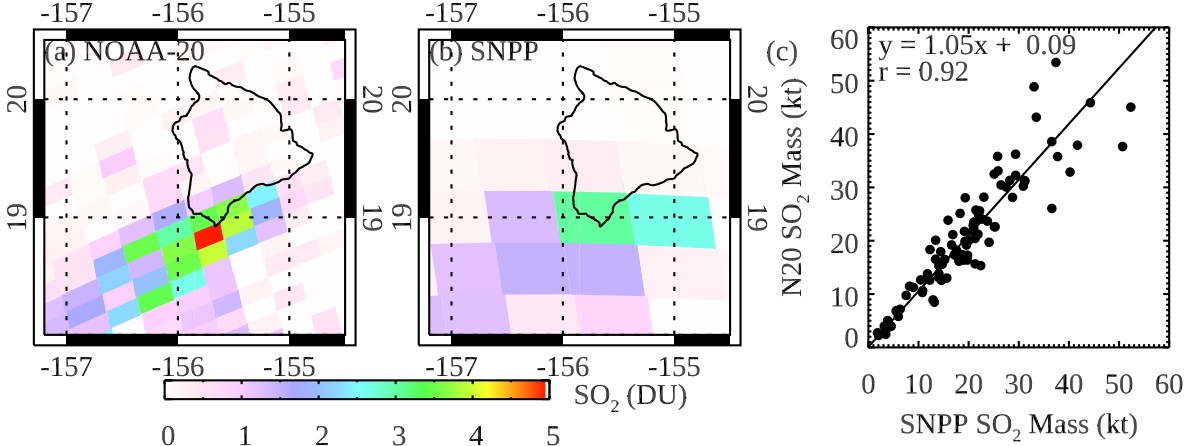

**Figure 10: Volcanic SO₂ VCDs around the Kilauea volcano in Hawaii on 8 May 2018 retrieved from (a) N20/OMPS and (b) SNPP/OMPS, assuming an *a priori* profile with a center mass altitude of 3 km (lower troposphere or TRL). (c) The scatter plot of N20 *vs.* SNPP daily SO₂ mass near Kilauea during May to June 2018. The best fit line from a regression analysis between the two**

**daily SO₂ mass datasets, along with its slope and intercept, is also given in (c). For the calculation of daily SO₂ mass, TRL volcanic SO₂ VCDs are first gridded to 0.5°×0.5° resolution using all pixels with SZA < 70° for both instruments. The daily mass within the domain (as in Fig. 10a and 10b) is then calculated by taking the sum of SO₂ mass from all grid cells that have gridded SO₂ VCD > 0.1 DU. Changing the threshold to 0.2 DU yields similar correlation coefficient (r = 0.92) and slope (1.04).**

The other case for comparison is the Raikoke eruption on 21 June 2019. The explosive eruption injected sizable amounts of

SO₂, ash, and sulfate aerosols into the lower stratosphere. The SO₂ plume, soon dispersed over much of the northern hemisphere, could be observed from satellite instruments several weeks after the eruption (Gorkavyi et al., 2021). Here, we estimate SO₂ mass within the Raikoke plume for each day after the eruption until the end of July 2019, using STL (lower stratosphere, the assumed center altitude of the plume is 18 km) SO₂ VCDs from N20 and SNPP/OMPS. The SO₂ mass estimates from the two products (Fig. 11) are well correlated ($r = 0.98$) and agree to better than ±15% for all days except for

21-22 June 2019 and 28-31 July 2019. Immediately following the eruption, high SO₂ concentrations in the dense volcanic plume can saturate the SO₂ absorption signals at shorter wavelengths, resulting in low biases in SO₂ retrieved from UV instruments (see Li et al., 2017 for more details). Moreover, the effects of the light-absorbing volcanic ash are not explicitly accounted for in the current OMPS volcanic SO₂ algorithm and may cause additional low biases in SO₂ retrievals. These two factors likely have different impacts on retrievals from N20 and SNPP/OMPS, thus leading to relatively large differences

between the two instruments at the beginning of the time series (Fig. 11). As for 28-31 July 2019, with the significant drop in



SO₂ VCDs due to plume dispersion and chemical loss, mass estimates are likely more strongly influenced by areas outside of the actual plume. The relatively large differences between N20 and SNPP/OMPS may point to slightly larger background noise in N20/OMPS retrievals, especially at higher latitudes.

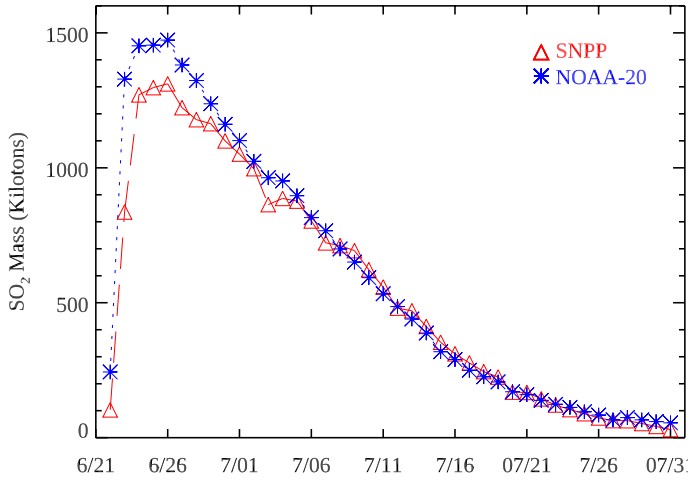

**Figure 11: The N20 and SNPP times series of daily SO₂ mass within the Raikoke volcanic plume after the eruption in June 2019, based on STL (18-km profile) volcanic SO₂ retrievals. The STL SO₂ VCDs from both instruments are first gridded to 0.5°×0.5° resolution using the same process as for the Kilauea case study. To minimize the impact of other SO₂ emission sources, we only count grid cells located north of 20°N that have VCDs > 0.2 DU in the estimates of daily Raikoke SO₂ mass.**

## 4 Data Availability

The version 1 NOAA-20/OMPS PCA SO₂ product is available at NASA's Goddard Earth Sciences Data and Information Services Center (GES DISC):

https://disc.gsfc.nasa.gov/datasets/OMPS_N20_NMSO2_PCA_L2_Step1_1/summary (last access: 13 May 2024). The DOI identifier is https://doi.org/10.5067/OMPS/OMPS_N20_NMSO2_PCA_L2_Step1.1 (last access: 13 May 2024, Li et al., 2023).

The version 2 SO₂ emission catalogue based on OMI, TROPOMI, SNPP, and N20 data is available at GES DISC:

https://disc.gsfc.nasa.gov/datasets/MSAQSO2L4_2/summary (last access: 13 May 2024). The DOI identifier is https://doi.org/10.5067/MEASURES/SO2/DATA406 (last access: 13 May 2024, Fioletov et al., 2022).



# 5 Conclusions

In this paper, we introduce our newly released version 1 NOAA-20/OMPS SO$_2$ product. Generated with our PCA-based
retrieval algorithm that is also used for NASA standard OMI and SNPP/OMPS SO$_2$ datasets, the N20/OMPS SO$_2$ product aims
to extend NASA's long-term global SO$_2$ data record into the JPSS era. The N20/OMPS SO$_2$ retrieval algorithm shares many
similarities with the OMI and SNPP/OMPS algorithms. This is especially the case for volcanic SO$_2$ retrievals, and comparisons
for the 2018 Kilauea eruption and 2019 Raikoke eruption demonstrate good consistency between N20 and SNPP/OMPS. On
the other hand, several modifications have been made for anthropogenic (or PBL) SO$_2$ retrievals, considering the instrumental
characteristics of N20/OMPS Nadir Mapper (*i.e.*, greater spatial resolution but reduced signal-to-noise ratio for N20 *vs.*
SNPP/OMPS) as well as the availability of other input datasets (*e.g.*, the Raman cloud product).

Statistical analyses of SO$_2$ SCDs confirm that N20/OMPS, like its predecessor SNPP/OMPS, can produce good quality
retrievals suitable for long-term global monitoring of large anthropogenic sources and degassing volcanoes. In the absence of
significant volcanic plumes, both N20 and SNPP/OMPS retrievals show generally small biases, with mean SCDs within ±0.05
DU over the remote Pacific. At its native resolution, the standard deviation of N20/OMPS SO$_2$ SCDs over the same areas is
~0.35-0.6 DU and 2-4 times larger than SNPP/OMPS. Once aggregated to the SNPP/OMPS resolution, the N20/OMPS SCDs
have comparable standard deviation except at high latitudes (70-80°N). This suggests that the greater noise in N20/OMPS SO$_2$
retrievals is largely driven by reduced SNRs in measurements at higher spatial resolution. Retrievals at large solar zenith angles
are probably subject to additional errors. Both N20 and SNPP/OMPS SO$_2$ SCDs demonstrate remarkable stability over the
entirety of their missions thus far. There are no significant changes in the mean SCDs over the equatorial Pacific and very
small trends in the standard deviation of SCDs (0.0002 DU y$^{-1}$ for SNPP/OMPS, and -0.00023 DU y$^{-1}$ for N20/OMPS at SNPP
resolution).

Comparisons of N20 and SNPP/OMPS PBL SO$_2$ VCDs, calculated using SCDs and a fixed AMF of 0.36, show similar spatial
distribution between the two retrievals, with both revealing SO$_2$ signals over major polluted areas such as India, the Middle
East, and South Africa as well as degassing volcanoes. N20/OMPS PBL SO$_2$ VCD is slightly lower over most areas as
compared with SNPP/OMPS, outside of high latitudes and SAA-affected areas. The causes for these differences are not fully
understood at this point but are at least partially attributable to algorithmic differences for SCDs, including algorithm settings
(*e.g.,* the threshold for filtering out SO$_2$-laden pixels) and a noise reduction scheme for SAA areas that is implemented for
SNPP but not yet for N20/OMPS. Even though the overall negative biases in N20/OMPS retrievals are quite small, they lead
to substantial differences in the long-term PBL SO$_2$ time series between the two OMPS datasets. Over selected anthropogenic
source areas, the N20 and SNPP/OMPS time series are well correlated (r > 0.8 for all but one areas) but the SO$_2$ mass based
on N20/OMPS is on average ~20% smaller over the Middle East and India. Tighter correlation (r > 0.9) and better agreement
(typical differences of ~10-15%) are found for selected degassing volcanoes, implying that sources with strong SO$_2$ signals
are less susceptible to the algorithmic differences.



Despite the negative biases in N20/OMPS retrievals as compared with SNPP/OMPS, top-down SO$_2$ emission estimates for large point sources (included in the version 2 emission catalogue) show very good agreement between N20/OMPS and other instruments including OMI, SNPP/OMPS, and TROPOMI ($r$ > 0.98, differences in total emissions < 10%). To assess the sensitivity of each instrument to SO$_2$ sources, we also compute the ratios between the estimated emissions and their uncertainties. As expected, TROPOMI has the highest ratios owing to its high spatial resolution and relatively small retrieval

noise; the ratios for N20/OMPS are slightly larger than OMI and greater than SNPP/OMPS for most sources, suggesting that of the two OMPS instruments, N20/OMPS is better suited for continuing the long-term emission catalogue started with OMI. It is also interesting that for the Middle East and India, emission estimates from N20/OMPS are smaller than those from OMI and TROPOMI, whereas the emissions based on SNPP/OMPS are greater. This suggests that the relatively large differences in the PBL SO$_2$ over these two regions could reflect the combined effects of the negative and positive retrieval biases from

N20 and SNPP/OMPS, respectively.

In summary, through extensive evaluation of the version 1 N20/OMPS SO$_2$ product and the version 2 emission catalogue, we have demonstrated that N20/OMPS Nadir Mapper can further extend the long-term NASA SO$_2$ climate data records from OMI and SNPP/OMPS. Efforts are currently underway to enhance the consistency between the next version N20/OMPS SO$_2$ product and the OMI and SNPP/OMPS products, including refinement of the N20/OMPS SO$_2$ SCD retrievals as well as

development of a N20/OMPS Raman cloud product that will be used for AMF/Jacobian calculations. These efforts, along with the recent advances in machine learning techniques for reducing retrieval noise and biases (e.g., Joiner et al., 2023; Li et al., 2022), will facilitate the development of multi-decade, coherent global SO$_2$ datasets across multiple satellite missions.

**Author Contribution:**

CL designed and implemented the retrieval algorithm, performed tests, and prepared the manuscript. NK, SC, JJ, and AV made

suggestions on the design of the retrieval algorithm. PL assisted in the production code for the retrieval algorithm and the public release of the emission catalogue. VF, CM, and DG generated emission estimates for the catalogue. JJ first proposed PCA-based trace gas retrieval technique. CS provided L1B and total O$_3$ input data for the SO$_2$ retrieval algorithm. All authors commented on the manuscript.

**Competing Interests:**

The authors declare that they have no conflict of interest.



**Acknowledgements**

We would like to thank the NASA Suomi National Polar-Orbiting Partnership (NPP) and the Joint Polar Satellite System (JPSS) Satellites Standard Products for Earth System Data Records program for funding of the SNPP and N20/OMPS $SO_2$ product development and analysis (Grant # 80NSSC22K0158). The satellite based $SO_2$ emission catalogue is partially
supported by NASA Making Earth System Data Records for Use in Research Environments (MEaSUREs) program. We also thank the NASA Ozone Processing Team (OPT) for the L1B and total $O_3$ data from SNPP and N20/OMPS.

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
