# Peer review of "Version 1 NOAA-20/OMPS Nadir Mapper Total Column SO2 Product: Continuation of NASA Long-term Global Data Record"

_Earth System Science Data, 2024_

## Author Comment (AC1)

**Response to Referee Comment 1 (RC1) from Anonymous Referee # 1** (*Referee comment in Italic*, response in blue).

*RC1: 'Comment on essd-2024-168', Anonymous Referee #1*
*In their paper Can Li and co-authors present the NOAA-20/OMPS SO2 product. The paper is well written. It has a good introduction with a well chosen set of references. The reason for the work, linked to the creation of a multi-sensor long-term coherent NASA/NOAA SO2 data record, is motivated extensively (maybe even more than actually needed). The new dataset is complementing and extending the series from OMI and SNPP/OMPS, and it is important that this dataset is documented in the peer-reviewed literature. The added value of N20 compared to the other platforms is presented in a nice way. It was good to see the emission estimate comparisons, which are very convincing. I am in favour of publishing this work after my comments have been addressed.*

We thank the referee for the thorough review. We have carefully considered all the comments and updated the manuscript accordingly. Please find below our point-to-point response.

**General comments:**

*Validation (e.g. with MAXDOAS, PANDORA, IR satellite data) is not discussed in the paper. Could you add one or two paragraphs on how the (previous OMI/SNPP) retrieval compare with independent observations?*

In the revised manuscript, we have added a paragraph on previous studies that compare OMI/OMPS PCA SO$_2$ retrievals with other ground-based and satellite datasets.

*It seems that the product does not contain averaging kernels, like most recent satellite data products do. This would be useful, in particular for the PBL contributions and for data assimilation applications. Is there a chance that kernels will be added in a coming upgrade?*

Yes, for the next version (version 2) N20/OMPS SO$_2$ product, we plan to use a Raman cloud product that is currently under development (along with model-based *a priori* profiles) as input to calculate (vertically resolved) box AMFs for individual pixels. The box AMFs will be included in the level 2 output files for the v2 product. We have clarified this in the final summary paragraph of the revised manuscript.

*The retrieval is called "version 1", but there are still a few elements missing, in particular the cloud retrieval, AMF calculation and the SAA filtering. Is a new update (v1.1) foreseen for the near future to include these aspects? The final summary paragraph mentions further extensions using machine learning, but this sounds like a more longer term development path (v2.0).*

As mentioned above, we have clarified in the final summary paragraph that we intend to implement these updates for the next version N20/OMPS SO$_2$ product. Please note that the same updates have already been implemented with our latest (v2) OMI and SNPP/OMPS SO$_2$ products and we plan to designate this next version N20/OMPS product as v2, to keep with our OMI and SNPP product naming convention.

Indeed, the machine learning techniques are part of a longer-term development path, and we plan to conduct more research before implementing them in our standard data production pipeline.

***Detailed comments:***

*l 125: "none-SO2 PCs". Should this be "non-SO2"?*

Corrected.

*l 141: "Process each orbit separately". If an orbit contains mainly ocean, is there still enough information stored in the PCA to descibe pixels over land? Isn't it better to train the PCA with one or multiple days of data?*

We previously tested different ways to conduct PCA, including using data from one day or multiple days and found no obvious improvements over training using a single orbit. We found that by processing each orbit separately, the algorithm can better account for orbit-to-orbit changes in the measurements (such as dark current). This also simplifies the setup of the algorithm in an operational processing environment, including for real-time (direct readout) data production. As for the referee's question regarding land vs. ocean, we don't expect it to be a major source of errors, given that the spectra are typically well-characterized by the PCs (with > 99.9999% of variance explained by the ~20 leading PCs). That said, we think it could be useful to test if better retrievals can be achieved by stratifying the data for PCA (for example based on surface type, elevation, and cloudiness etc.). This is beyond the scope of the current data description paper but would be an interesting topic for a sperate study.

*l 183: (new volcanic screening): What is the impact? are there now more, or less pixels flagged?*

As shown in Figure 1 below, the new volcanic SO$_2$ flagging scheme is more sensitive and flags more pixels as compared with the old scheme. For NOAA-20/OMPS, the new scheme flags pixels with STL SO$_2$ of ~2 DU and above, whereas the old scheme flags pixels with STL SO$_2$ of ~5 DU and above. We have added this information to the revised manuscript and the figure to the supplemental information (Fig. S1).

[Figure]

Figure 1. NOAA-20/OMPS pixels flagged for potential volcanic $SO_2$ signals from the Ruang volcanic plume on 30 April 2024 using (a) the new and (b) the old volcanic $SO_2$ flagging scheme. (c) $SO_2$ vertical column densities (VCDs) retrieved assuming a plume height of 18 km.

*l 194: "five subsectors". Are these five latitude bands? Why is this important? Please add a few lines to explain the subsectors.*

We have clarified in the paper that these bands are based on latitudes and solar zenith angles. We previously found that they help to reduce retrieval biases. In addition, we have conducted a test without using the subsectors and confirmed that for NOAA-20, the use of subsector does reduce retrieval biases (see Figure 2 below). The figure and relevant discussion have been added in the supplemental information (Fig. S4) and section 3.1 of the paper, respectively.

[Figure]

Figure 2. Same as Fig. 2a in the paper manuscript but with N20/OMPS retrievals conducted without subsectors.

*l 202: "has not yet been implemented with N20/OMPS." Is there a special (technical) reason why this has not yet been implemented? The benefits seem to be big, as shown in Fig. 4.*

The scheme for noise reduction over the SAA areas has been previously implemented with OMI and SNPP/OMPS as part of the algorithm that also calculates AMF/Jacobians for anthropogenic SO$_2$ for each pixel. As noted by the referee, that part of the algorithm also requires Raman cloud retrievals as input, and thus cannot yet be implemented with N20/OMPS. To maximize the use of a common code base between different instruments, we elected to defer the implementation of the noise reduction scheme with N20/OMPS.

*l 203: "a fixed AMF (0.36) is used to convert all SCDs to SO2 VCDs," I was confused here, because line 169 mentioned "and a priori SO2 profiles based on a climatology from multi-year global model simulations". Please clarify. I assume this is linked to the missing cloud product. Maybe move this remark up to the beginning of the paragraph.*

Yes, the fixed AMF is used in version 1 N20 product due to the lack of the cloud product, which is currently under development. We have changed the text in the manuscript accordingly.

*l 248: "the the"*

Corrected.

*Fig.2 left seems to indicate jumps in the NH. Is this within the noise, or linked somehow to the subsectors?*

As mentioned above, we conducted a test without using the subsectors and the results suggest that the use of subsectors helps to reduce retrieval biases (and jumps).

*The biases seem to be linked to changes in the algorithm, and especially the threshold for SO2 containing pixels. In Figs 6 and 7 the product is again bias corrected to check consistency with SNPP. This leaves the question why the procedure was changed in N20 after all. Why do you bias-correct N20 and not SNPP? It would be useful to extend the discussion on this. Does the old procedure lead to problems for N20? If so, where do the problems occur and how big are these problems?*

We previously tested the OMI/OMPS SO$_2$ algorithm with N20/OMPS with minimal changes in algorithm settings and noticed a substantial positive bias over background areas (see Figure 3 below) that is not noticeable in our OMI or SNPP/OMPS retrievals. To reduce this bias, we updated several algorithm settings, including the threshold for

SO$_2$ containing pixels, with N20/OMPS (see Figure 3b below).  We have added the figure to the supplemental information (Fig. S3).

As for using SNPP/OMPS as the reference to bias-correct N20, the choice is made based on 1) the longer and more established data record from SNPP; 2) relatively good consistency between SNPP/OMPS and OMI (e.g., Li et al., 2017; Zhang et al., 2017), and 3) the fact that substantial calibration work is still ongoing for N20/OMPS. We have added this explanation to the revised manuscript.

[Figure]

Figure 3. SO$_2$ vertical column densities (VCDs) for April 1, 2020, retrieved from N20/OMPS using (a) the same algorithm settings as for SNPP/OMPS, and (b) the updated algorithm settings. A constant air mass factor (AMF) of 0.36 is applied to all pixels in both retrievals.

*Fig.8: It is nice to see the N20/SNPP ratio plots. Indeed, this seems to indicate that the increase in resolution has a benefit over the lower SNR for emission estimates.*

*What I found surprising is the good match with TROPOMI. The retrieval is very different, the AMF is different and the SNR and resolution are also very different. It would be useful to discuss this in more detail, maybe even add an extra figure, and an extra paragraph. Apparently the local contrasts in SO2 columns and absolute column values are very similar compared to the NASA algorithm.*

For TROPOMI, the SO$_2$ data used in emission estimates are based on the COBRA algorithm that is conceptually similar to the PCA algorithm. Previous comparisons (Theys et al., 2021) indicate largely consistent SO$_2$ SCDs between TROPOMI COBRA and OMPS PCA retrievals. The local bias corrections in the emission estimates further reduce the differences between instruments/algorithms.

While differences exist in how AMFs are calculated for different instruments/algorithms, for emission estimates, the same set of location-specific AMFs

are applied to different satellite datasets, thus eliminating the AMF as a source of differences.

We have added the discussion above to the revised manuscript.

*l 448: A weighted average based on inverse variances is suitable for unbiased datasets. In the average of the four platforms algorithm and instrument differences play a role. Why did the authors choose this approach?*

Please note that the emission estimation algorithm removes local biases, so we are averaging unbiased datasets even if the original satellite $SO_2$ data have some biases.

This inverse variances-based approach was introduced in Version 2 of the $SO_2$ emissions catalogue (https://essd.copernicus.org/articles/15/75/2023/) when it was necessary to combine emission estimates from several satellite instruments with very different emission uncertainties.

The contribution of individual satellite sources as a function of the source emission strength is shown in the figure below (Figure 4). For small sources (under 30 kt/year), only TROPOMI was able to produce emission estimates with low uncertainties. For large sources, all satellite instruments can be used for emission estimates. So, in the former case, the weighted average was based mostly on TROPOMI data, while in the latter case, all satellite instruments provided comparable contributions. NOAA-20/OMPS based emission estimates have slightly smaller uncertainties compared to OMI, as also reflected by the weighting coefficients. The Figure below has been added to the supplemental information (Fig. S7)

[Figure]

Figure 4. Relative contribution of individual satellite instruments to the weighted average for emissions estimate depending on the emission strength for 2018-2023.

*Sec 3.5: It would be interesting to see the TROPOMI results as well for the two eruptions.*

We have added TROPOMI results for both eruptions.

*Inviting one representative from the TROPOMI retrieval team as co-author could be considered.*

We considered this and elected to not invite the TROPOMI retrieval team as the initial manuscript was heavily focused on OMPS. For the revised manuscript, more publicly available TROPOMI data are used. We feel that it is appropriate to acknowledge the TROPOMI retrieval team for the use of those data.

*The code and data availability section is missing.*

We use the template from the journal website and the data availability is in section 4. We will check with the editorial office to verify where the section should be in the final version of the paper.
For code availability, we will check with the relevant NASA authorities on requirements for software release.

---

## Author Comment (AC2)

**Response to Referee Comment 2 (RC2) from Anonymous Referee # 2** (*Referee comment in Italic*, response in blue).

**RC2**: 'Comment on essd-2024-168', Anonymous Referee #2
**Review of the manuscript "Version 1 NOAA-20/OMPS Nadir Mapper Total Column SO2 Product: Continuation of NASA Long-term Global Data Record" by Can Li et al.**

*The manuscript entitled "Version 1 NOAA-20/OMPS Nadir Mapper Total Column SO2 Product: Continuation of NASA Long-term Global Data Record" describes the NOAA-20 (N20)/OMPS SO2 product, which aims at extending the long-term climate data record of OMI, SNPP/OMPS of SO2 column densities from both anthropogenic and volcanic activities.*

*The authors not only describe the new algorithm for N20/OMPS but also perform a comparison with the existing data record, showing the added value of this additional satellite product.*

*The manuscript is very well written and already in a very good state and require only minor revision, as detailed in the detailed comments hereafter:*

We thank the referee for the detailed comments. We have carefully considered all suggestions and made changes to the manuscript accordingly. Below please find our point-to-point response.

*Detailed comments:*

*Figure 1: Instead of showing arrows for each instrument that all end in 2024 it is perhaps better to show the planned mission timeline. Otherwise one gets the impression that all missions end in 2024 (except for JPSS3-4/OMPS). What is meant with "Direct readout only" for NOAA-21/OMPS?*

Other than OMI, we are not aware of any planned ending dates for other missions. We have updated Figure 1 to clarify this.

Direct readout SO$_2$ retrievals (with limited areal coverage) from NOAA-21/OMPS are currently available from few ground stations that have the hardware and software to receive and process broadcast data from NOAA-21. This is for producing real-time SO$_2$ data (latency < 30 min from satellite overpass) for monitoring and mitigating volcanic hazards for aviation. We have clarified this point in the figure and added a reference for this application.

*Section 2.2.1 Line 134: Perhaps mention which RT code is used for the calculation of AMFs*

We use VLIDORT for RT calculations. This information has been added to the revised manuscript.

*Section 2.2.1 Line 162: You mention that the pixels are subdivided into 3 subgroups based on their latitude. What is the latitude range of each subgroup?*

In the revised manuscript, we have clarified that the subgroups (subsectors) are based on latitudes and solar zenith angles (SZAs) and provided details on how the threshold is calculated.

*Section 2.2.2 Line 185: You are using two reference orbits to derive the PCs. Can you show or indicate the effect on your results when you use a different orbit/day, e.g. in 2024?*

We expect that different reference PCs will lead to changes in the initial $SO_2$ estimates, and as a result, different pixels getting selected for PCA and consequently different final estimates of SCDs. We have conducted test retrievals for 1 April 2023 using two different sets of reference PCs (see Figure 1 below). For most pixels, we found minor differences in SCDs that are well within the typical retrieval noise (mean difference of ~0 DU, and standard deviation of the differences < 0.1 DU). There are larger differences for orbits that pass over SAA areas, indicating that retrievals for those orbits are more sensitive to initial $SO_2$ estimates owing to larger retrieval noise due to SAA. We have added this discussion to the revised manuscript (and figure to the supplemental information, Fig. S2).

[Figure]

Figure 1. The density map comparing the final N20/OMPS $SO_2$ SCD retrievals using reference PCs from orbit 17460 on 1 April 2021 *vs.* those using reference PCs from orbit 33010 on 1 April 2024. (a) includes all orbits on 1 April 2023, whereas (b) includes orbits on the same day that are unaffected by SAA.

*Section 2.3: I guess think this section should be moved to the end since it is out of context at this location and disturbs the readability.*

We have moved this section to the supplemental information.

*Section 2.4 Line 248. Typo "the the"*

Corrected.

*Section 3.1, Figure 2 and 3: From the figure you see an offset between the mean SO2 VCD of SNPP/OMPS and N20/OMPS. Where is this bias of SNPP/OMPS coming from? Maybe you should address this as well. Does a rebinning of N2O/OMPS have an effect on the mean SCD and associated bias?*

The offset between mean $SO_2$ from SNPP/OMPS and N20/OMPS is likely due to different algorithm settings, especially the threshold for pixels that are assumed to contain $SO_2$ and excluded from PCA analysis. We have added the discussion to the revised manuscript.

Binning N20/OMPS is not expected to significantly change the mean SCDs. Indeed, this is confirmed by the results shown in the figure below.

[Figure]

Figure 2. Same as Fig. 2a in the paper manuscript but with N20/OMPS mean $SO_2$ SCDs (blue) calculated after first binning the pixels to SNPP/OMPS resolution.

*Section 3.2 Figure 4: Why do you see a stronger difference in mountain areas, especially in the South American Andes (negative) and Scandinavia (positive difference). Is this related to icy surfaces and related albedo effects?*

In the paper, we have also noted larger differences over coastal areas (including Scandinavia). The reason for this is currently unknown. It is possible that there are terrain or surface related biases that are amplified in N20 retrievals due to its smaller

pixel size and the biases are not completely averaged out. This would be an interesting topic for future algorithm refinement studies.

*Section 3.3 Third paragraph & Figure 6: The differences in the text and in the figure subtitles are slightly different, probably due to different rounding. E.g. for Norilsk a 8% difference is written in the text, but the figure title states 7% difference.*

Yes, this is due to different rounding. We have updated the figure and the text so that the numbers are consistent.

*Section 3.3 Figure 6. It is really hard to distinguish the three colored lines from each other. Maybe it would help if you show only the timeframe with N20/OMPS results, i.e. show the plot with data from 2018 onwards?*

We elect to keep the time series unchanged in the paper, as there is also discussion on the long-term changes in SO$_2$ (e.g., over India and China). We have added a figure for the period of 2018-2023 (Fig. S5) to the supplemental information.

*Section 3.3 Figure 7 Same suggestion as the two above: Perhaps show only data for 2018+ and check numbers in text and figure title.*

We have checked the numbers and added a time series figure for 2018-2023 (Fig. S6) to the supplemental information.

*Section 3.4 Figure 8. The x axis label of d and e are missing a "/" character, i.e. N20/OMPS instead of N20 OMPS.*

*Section 3.4 Figure 8d-f. Perhaps you find a better y axis label, since "OMI and SNPP ratio" is a bit hard to understand when only looking at the figure. Perhaps use "OMI emission/uncertainty ratio" or so.*

We have updated the axis labels for Fig. 8 accordingly.

*Section 3.5 Figure 10 Perhaps it would be useful to show the comparison with TROPOMI.*

We have added TROPOMI data to Figure 10 (and Figure 11).

*Section 4, This section should appear after the conclusions and then Section 2.3 should come after (see my comment above).*

We use the template provided by the journal and the data availability section comes before the summary. We will check with the editorial office regarding the order of sections.